# Identification of Tools for the Functional and Subjective Assessment of Patients in an Aquatic Environment: A Systematic Review

**DOI:** 10.3390/ijerph17165690

**Published:** 2020-08-06

**Authors:** Antonio Cuesta-Vargas, Jaime Martin-Martin, Manuel Gonzalez-Sanchez, Jose Antonio Merchan-Baeza, David Perez-Cruzado

**Affiliations:** 1Department of Physiotherapy, Faculty of Health Science, University of Malaga, 29071 Malaga, Spain; acuesta@uma.es (A.C.-V.); mgsa23@uma.es (M.G.-S.); 2Institute of Biomedical Research in Malaga (IBIMA), 29071 Malaga, Spain; josan_mb@hotmail.com (J.A.M.-B.); d_perez_cruzado@hotmail.com (D.P.-C.); 3Legal Medicine Area, Department of Human Anatomy, Legal Medicine and History of Science, Faculty of Medicine, University of Malaga, 29010 Malaga, Spain; 4Research group on Methodology, Methods, Models and Outcomes of Health and Social Sciences (M3O), Faculty of Health Science and Welfare, University of Vic-Central University of Catalonia (UVIC-UCC), 08500 Vic, Spain; 5Department of Occupational Therapy, Universidad Católica San Antonio de Murcia, 30107 Murcia, Spain

**Keywords:** functional test, questionnaires, aquatic therapy, assessment

## Abstract

Aquatic therapy is one of the most common treatments for alleviating musculoskeletal pathologies. Its effectiveness has been evaluated with functional tests and questionnaires. Functional tests are used in aquatic therapy; however, in most cases, they are carried out in a non-aquatic environment and, as such, their results may differ from those of tests performed in an aquatic environment. A systematic review was performed to assess the accuracy of functional tests and patient-reported outcomes to assess aquatic therapy interventions. The authors conducted a literature search in July 2019. In total, 70,863 records were identified after duplicates removed. Of these, 14 records were included about functional tests assessment in aquatic environment and 725 records for questionnaires. The majority of the tests had also been assessed in a dry environment, allowing differences and similarities between the tests in the two environments to be observed. Different variables have been assessed in tests included in the present systematic review (cardiorespiratory, neuromuscular, kinematic, physiological, kinetic responses and rating of perceived exertion) which are included in the manuscript. Visual Analogue Scale, Western Ontario and McMaster Universities Osteoarthritis Index and the 12-item Short Form Health Survey were the assessments most commonly used by the different authors.

## 1. Introduction

Hydrotherapy has been used to describe a wide range of activities, most of which are therapeutic activities and activities in heated swimming pools. In recent years, it has come to be used by rehabilitation clinicians due to its many benefits in a wide range of different pathologies [1].

The physical properties of hydrodynamics, such as buoyancy, viscosity and thermodynamics, appear to improve mobility in populations with disabilities [2]. As a result of these physical properties of water, therapy in this environment facilitates muscular strengthening as well as aerobic exercise in certain populations that suffer serious pain or experience great difficulty in performing exercises on hard surfaces, while at the same time also helping to reduce the risk of any type of injury [3]. In addition, aspects such as the great social support that comes with the exercise environment, as well as water temperature, help to improve other psychosocial variables such as fatigue, anxiety and depression [4].

Aquatic physical therapy treatment (individual or group) should incorporate individual assessment, diagnosis and the use of clinical reasoning skills to formulate an appropriate treatment program for the patient [5]. The use of functional tests is very beneficial when carrying out any type of therapy, since they provide us with relevant information on which to base clinical decisions or evaluate patients receiving the therapy [6,7]. On the other hand, patient-reported outcomes provide clinicians with important information about different constructs (pain, quality of life and activities of daily living) in order to understand if the aquatic therapy can improve these variables [8,9].

When evaluating a patient, it is not enough to have information provided by the patient or for the evaluation to be performed on a dry surface. Both dry-land and aquatic assessments are necessary to ensure that the right clinical decisions are made [10]. Functional tests are used in aquatic therapy; however, in most cases, they are carried out in a non-aquatic environment and, as such, their results may differ from those of tests performed in an aquatic environment [11,12]. In contrast, questionnaires used in aquatic therapy provide the same information in both environments, but there are no systematic reviews on the use of self-reported assessments of aquatic therapy.

Therefore, the objective of the present systematic review was to analyze the outcome measures of functional tests and patient-reported outcomes used within an aquatic environment to identify those with a greater degree of applicability in this environment.

## 2. Material and Methods

A systematic review was carried out to identify, appraise and synthesize the functional tests that have been used in aquatic environments in line with Cochrane criteria for systematic reviews. The protocol of this systematic review was developed and has been registered in the International Prospective Register of Systematic Reviews (PROSPERO CRD42017071068). The Preferred Reporting Items for Systematic Reviews and Meta-Analyses (PRISMA) guidelines were used for designing the present study [13].

### 2.1. Data Sources and Searches

To carry out the present systematic review, the authors conducted a literature search in February 2020 using the PubMed, Scopus, Scielo and Otseeker databases. The search string used contained the following keywords: “water” OR “aquatic therapy” OR “hydrotherapy” AND “functional test” OR “test” OR “functional assessment” OR “functional” OR “assessment” OR “questionnaire” OR “report” OR “scale”. No time limit was set for the beginning of the search and all types of studies involving a functional test or patient-reported outcomes in an aquatic environment were included. The studies chosen for inclusion were independently selected, blindly, by two researchers (DPC and JMM) in two stages: in the first stage, the researchers considered the title and abstract of the selected studies; and, in the second, they evaluated the full text of the remaining articles. Both reviewers then discussed any discrepancies, which were resolved after reading the full text again and further discussion. Where discrepancies between the two authors remained, a third reviewer (MGS) assessed the articles’ eligibility.

### 2.2. Study Selection

For functional tests, the studies included were those in which functional tests had been conducted in an aquatic environment in any type of population and their feasibility assessed, with a cut-off point of 15 points on the Stard scale [14]. The studies had to be published in English, Spanish, French, Portuguese or Italian.

For patient-reported outcomes, clinical trials were included that used therapeutic aquatic exercise (active exercise) as rehabilitation for any type of pathology, provided the studies included self-reported assessments using an instrument or tool among their results. Patient-reported outcomes used in these studies were extracted, and the outcomes that were used in five or more studies were extracted for analysis in the present systematic review.

### 2.3. Data Extraction and Quality Assessment

From functional tests, an analysis of the internal validity of the papers was carried out using the Stard scale for functional tests [14].

The Stard scale contains 34 items that evaluate different methodological aspects throughout an entire manuscript. The evaluation process consisted of two independent evaluators (DPC and JMM) who evaluated the studies and reached a consensus on the final score. A third reviewer (MGS) was required to confirm the selection of two of the articles included.

### 2.4. Data Synthesis and Analysis

The authors, the functions assessed, the environment in which the functions were assessed, the variables evaluated, the criterion validity and the reliability of the assessment were extracted from functional tests studies.

For patient-reported outcomes, the authors and the patient-reported outcomes assessed in each study were extracted, with the name, acronym, number of items, categories, time to administer, item rated, cost, cut-off, inter-rater reliability and intra-rater reliability, and the Cronbach’s alpha was used for internal consistency, convergent criteria validity, number of factors, sensibility/specificity, standard error of measurement and minimum detectable change shown for each outcome.

## 3. Results

In total, 101,257 studies were identified after the research strategy. For functional tests, 70,827 were eliminated after reviewing the title and abstract as they were found not to be related to the subject of study. On reviewing the full text of the remaining 36 articles, a further 19 were eliminated as they contained no evaluation of a functional test in an aquatic environment. The 17 selected articles were then put through an internal validity analysis using the Stard Checklist. A further three studies were subsequently excluded for failing to reach the cut-off of 15 points. As a result, 14 articles were included in the present systematic review of functional test assessment in aquatic therapy (see Figure 1).

For patient-reported outcomes, 53,754 were eliminated after reviewing the title and abstract. On reviewing the type of articles, 1716 articles were selected. Finally, 725 articles were included to extract the patient-reported outcomes (see Figure 2).

Appendix A (Appendix A) shows the scores assigned to the studies selected after the internal validity analysis carried out using the Stard Checklist. In this analysis, 17 papers that had passed the previous filters were included. The authors established a cut-off of 15 points to decide which items to include in the review. After analyzing each article and obtaining a final score, 14 of the 17 articles were included.

Appendix A (Appendix A) shows the characteristics of the studies included in the systematic review. It shows that all the functional tests were conducted in an aquatic environment, and that 11 were also carried out in a dry environment. In addition, the psychometric properties of the tests were assessed, being in this case the validity of the criterion and the reliability of said test being evaluated in all cases: cardiorespiratory, neuromuscular, kinetic, kinematic, physiological and rating of perceived exertion responses. The results of this systematic review show the functional tests that can be used for evaluation in aquatic therapy. Functional tests that have proven useful to the assessment in an aquatic environment are walking in water, deep water running, shoulder scaption, sit to stand test, time get up and go and vertical jump. They also show how important it is to evaluate such tests in an aquatic environment, since in most cases test responses will differ when performed in a dry environment.

Structural characteristics of the patient-reported outcomes used in an aquatic environment and included in the systematic review are shown in Appendix A (Appendix A). The 25 questionnaires were applied to compare the interventions carried out in dry and aquatic environments. The structural characteristics of the questionnaires were identified as number of items in each questionnaire, evaluation objective, average application time, scoring system, cut-off point and cost.

The psychometric properties of the questionnaires allow the assessment instrument to be chosen based on objective criteria. The properties identified were: reliability, internal consistency (Cronbach’s alpha), construct validity, factor analysis, sensitivity/specificity, standard error of measurement (SEM) and minimal detectable change. Appendix A (Appendix A) shows the psychometric properties of the questionnaires included in the systematic review.

## 4. Discussion

In the present systematic review, several functional tests were included that had all been conducted in an aquatic environment. The majority of the tests had also been assessed in a dry environment, allowing differences and similarities between the tests in the two environments to be observed. The exceptions were the studies by Cuesta-Vargas et al. [15] and Gayda et al. [16], which only evaluated tests in an aquatic environment. In addition, the articles included in this systematic review examined similar variables: cardiorespiratory, neuromuscular, kinetic, kinematic and physiological responses.

The results obtained in this systematic review and its application in the clinical setting should be specifically analyzed. The results of an objective test (functional tests) (Appendix A (Appendix A)) carried out in water and outdoors can offer different results. To these results, we must add the user’s own perception during the evaluation. The evaluation environment can offer greater security; however, a questionnaire (subjective variable) carried out in or out of the water will not change its result.

### 4.1. Walking

This test was conducted in five of the studies [17,18,19,20,21], in both an aquatic and a dry environment in all cases. In terms of differences in the neuromuscular response between the two environments, there were discrepancies between the different articles. The study by Alberton et al. [21] found no significant differences in this test between the two environments, nor did Masumoto et al. [20], whereas, in the study by Chevutschi et al. [17], significant differences were found in the erector spinae and soleus. Conversely, in the study by Alberton et al. [21], as with that by Masumoto et al. (18), significant differences were found in many muscle groups. Therefore, for this test, it seems that the neuromuscular test response differs depending on whether it is performed in or out of water.

Cardiorespiratory response was evaluated in the studies by Alberton et al. [21], Masumoto et al. [18] and Masumoto et al. [19]. Similarities were found between the Alberton et al. [21] and Masumoto et al. [19] studies, with both reporting significant differences between the two environments according to the rate of step at which the test was carried out. In contrast, in the study by Masumoto et al. [18], no significant differences were found between the three conditions in which the test was carried out. In this study, the test was not carried out at different rates, which may explain the lack of significant differences.

For variables such as kinetics, significant differences between the two environments and rates were found [21], while for physiological responses there were no significant differences between the three conditions in which the test was performed, for either systolic or diastolic pressure [18]. In the study by Masumoto et al. [19], which assessed rating of perceived exertion, again no significant differences between the two environments (water and dry) were found.

Finally, in the study by Alberton et al. [21], the reliability of this aquatic test was evaluated for neuromuscular response, with values ranging from 0.942 to 0.764 for intraclass correlation coefficient (ICC).

### 4.2. Deep Water Running

This test was evaluated in four studies [15,22,23,24] and was compared in all studies with a similar test in a dry environment.

The cardiorespiratory responses to this test were assessed in the studies by Gayda et al. [16] and Nagle et al. [25]. Here, discrepancies were found between both studies: the former found significant differences between environments, with higher values during the function of running on a static tape compared with the deep water running test, while the latter found significant correlations between the test performed in water and the function of running on a static tape. These differences could be explained by the fact that, in the study by Gayda et al. [16], the tests were carried out over a longer period of time and over three different protocols ((a) <8 min; (b) 8–12 min; and (c) >12 min) on the same day, while this diversity of protocols did not exist in the study by Nagle et al. [25], as the tests were performed over different days.

Physiological responses were evaluated in the studies by Masumoto et al. [26] and Cuesta-Vargas et al. [15]. Both found lower values of heart rate in the aquatic environment compared with the dry environment (treadmill). On the other hand, in the study by Masumoto et al. [26], neuromuscular responses showed less muscular activation when the test was conducted in water, while no significant differences for rating of perceived exertion were found between the two environments [water and dry).

For this test, the reliability of the cardiorespiratory response was also evaluated in the studies by Gayda et al. [16] and Nagle et al. [25], with values ranging of VCO_2_ = 0.92 [23] and RER = 0.65 [23].

### 4.3. Shoulder Scaption

This test was used in the study by Castillo-Lozano et al. [27] in both dry and aquatic environments and at different angles as well as at different speeds. Evaluating differences and similarities in neuromuscular responses, they found significant differences between the two environments in the pectoralis, middle deltoid and latissimus dorsi, in different planes and at different speeds.

### 4.4. Sit to Stand

This test was evaluated in dry and aquatic environments in the study by Cuesta-Vargas et al. [28,29]. These authors found significant differences in the neuromuscular response between the two environments in all the muscles evaluated.

### 4.5. Time Get Up and Go

This test was evaluated in dry and aquatic environments in the study by Cuesta-Vargas et al. [28], which revealed significant differences in neuromuscular response between the two environments in most of the muscles evaluated (rectus femoris, biceps femoris, tibialis anterior, soleus, gastrocnemius and erector spinae).

### 4.6. Jump

This test was evaluated in two studies [29,30], in both aquatic and dry environments. In both studies, concurrent kinetic responses were evaluated, and, in both cases, they were significantly higher in the aquatic environment than in the dry environment. However, Louder et al. [31] also found discrepancies: for certain movements, the speed and angle were higher in the aquatic environment, while, in other cases, they were higher in the dry environment (see Appendix A (Appendix A)).

### 4.7. Questionnaire Assessment

As mentioned above, a functional test can modify the results in and out of the water. However, a questionnaire that has subjective responses is not conditioned by the environment of completion. In this sense, the application of questionnaires for patient evaluation can be carried out based on the quality of life or on specific variables (functionality or impact of the pathology, among others) (see Appendix A (Appendix A)). The choice of questionnaire is influenced by different factors: structural characteristics, psychometric properties, duration of the questionnaire and scope of application.

The structural characteristics of the questionnaires determine the number of items, evaluation categories, application time, costs and scoring system (see Appendix A (Appendix A)). The assessment of quality of life by questionnaire can be performed with instruments such as the Short Form-12, Assessment of Quality of Life Scale and the Health Assessment Questionnaire, among others (see Appendix A (Appendix A)). Likewise, the direct impact of the disease on the person can be assessed by means of the Arthritis Self-Efficacy Scale, Berg Balance Scale, Chronic Venous Insufficiency Questionnaire or, to evaluate the functional capacity of the patient, the Assessment of Motor and Process Skills or the Physical Activity Scale for the Elderly, among other instruments (see Appendix A (Appendix A)). Therefore, the application of a set of questionnaires with chosen criteria provides a comprehensive view of the patient’s health status and will facilitate a therapeutic approach [32]. The application time of the instrument must be regulated; instruments such as the Child Health Assessment Questionnaire, Assessment of Motor and Process Skills or Short Form-36 (30–40 min) (Appendix A (Appendix A)) may be of excessive duration and can be difficult to integrate into an evaluation process within a clinical consultation. These assessments, despite their time of application, are tools with great acceptance in the scientific and clinical field, allowing reference values which enables comparison of results with other studies [33].

The psychometric properties of the questionnaires are fundamental as they allow us to identify the validity and reliability to apply the best instrument on each occasion [33] (Appendix A (Appendix A)). The highest test-retest reliability identified was in the Berg Balance Scale 0.99 [32] and the Chronic Venous Insufficiency Questionnaire 0.98 [34]. However, the lowest values were observed in the Assessment of Quality of Life Scale 0.78 [35]. Inter-observer reliability was only identified in two questionnaires: Falls Efficacy Scale; 0.72 [36] and Berg Balance Scale 0.98 [32]. The minimal detectable change of the included questionnaires was only identified in the Activities-specific Balance Confidence Scale [37], Arthritis Impact Measurement Scale [38] and the Berg Balance Scale [39]. Questionnaires such as the Activities-specific Balance Confidence Scale, Arthritis Impact Measurement Scale, Assessment of Motor and Process Skills, Berg Balance Scale, Child Health Assessment Questionnaire, Chronic Venous Insufficiency Questionnaire, Disease Activity Score, EuroQol-5 Dimension Questionnaire, Falls Efficacy Scale, Fibromyalgia Impact Questionnaire or GAD-10 have good internal consistency (≥0.9) (Appendix A (Appendix A)). Likewise, the instrument with the best sensitivity/specificity ratio of those reviewed was the Berg Balance Scale [40].

The choice of the questionnaire to apply will depend not only on the variable to be measured and whether it is validated but also on its applicability to the target population and its cross-cultural adaptation [41]. Cross-cultural adaptations must be validated so that the results obtained from the population analysis can be compared with other studies that use the same instrument [33]

This systematic review presents a series of limitations that must be taken into account when analyzing the results presented. One limit is that, although an attempt has been made to search databases with worldwide diffusion, there could be studies published in journals indexed in other databases not included in this systematic review. On the other hand, five languages were used in the selection of the documents (English, Spanish, French, Portuguese and Italian). There may be documents published in languages other than those indicated that have not been included in this systematic review.

## 5. Conclusions

The results of this systematic review show the functional tests that can be used for evaluation in aquatic therapy. They also show how important it is to evaluate such tests in an aquatic environment, since in most cases test responses will differ when performed in a dry environment. The present article is of vital importance to the clinical environment as well as the researcher, since it tells us which tests can be used for evaluation in an aquatic environment and allows us to begin evaluating other tests in this environment, which have, up until now, only been conducted in a dry environment.

## Figures and Tables

**Figure 1 ijerph-17-05690-f001:**
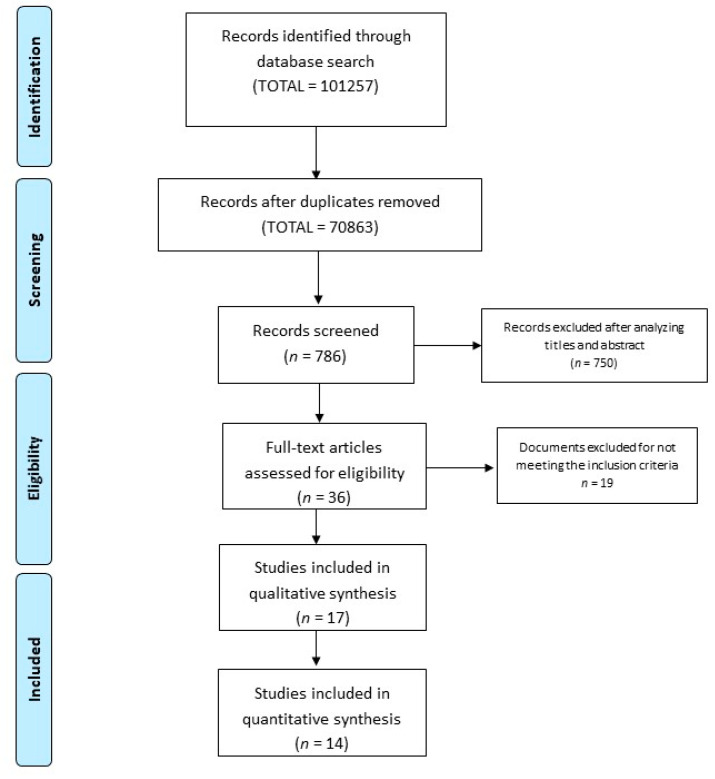
Study selection flow diagram.

**Figure 2 ijerph-17-05690-f002:**
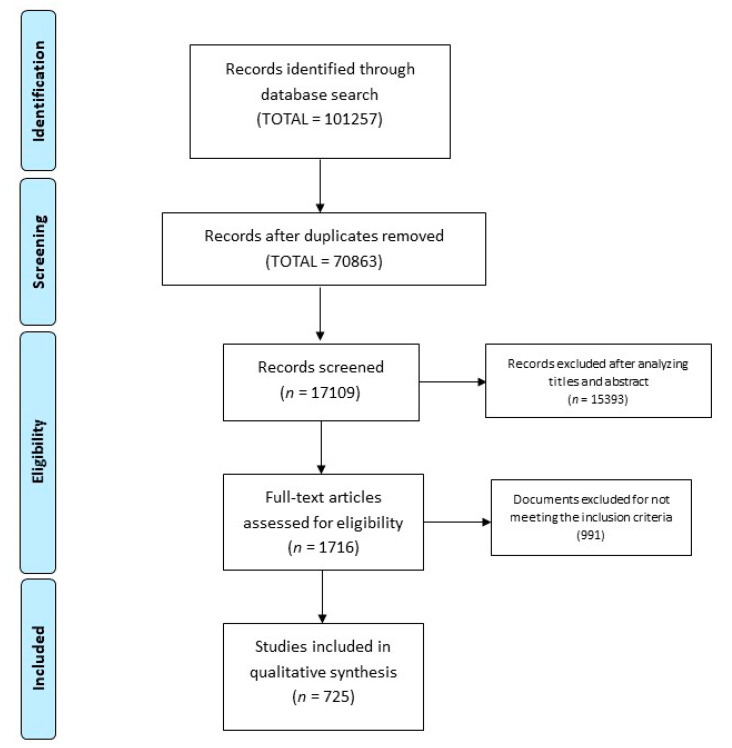
Study selection flow diagram.

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
