# Peer review of "Identification of Tools for the Functional and Subjective Assessment of Patients in an Aquatic Environment: A Systematic Review"

_ijerph, 2020, doi:10.3390/ijerph17165690_

Round 1

Reviewer 1 Report

Thank you for the opportunity to review this manuscript. In this paper, the authors presented systematic review was to analyse the outcome measures of functional tests and patient-reported outcomes used within an aquatic environment to identify those with a greater degree of applicability in this environment.

The manuscript is interesting and I believe that this manuscript presents novelty for publication.

In my opinion, this is a typical review using the appropriate methods for review.

if I can suggest changes, I suggest that Table 2. provide reference numbers in accordance with the references , it will help you find the right article.

If possible, it would be good to add numerical references to the articles in which the test was used in Table 3.

In the discussion, I also suggest using numerical references in accordance with the reference list, this will allow the reader to find the right article faster.

Author Response

ITEMIZED LIST OF THE REVIEWERS COMMENTS

Manuscript ID: ijerph-853695

Assistant Editor
Mrs. Adelina Platon

We would like to thank the Reviewers for their thoughtful and constructive comments. We have considered all suggestions and have incorporated them into the revised manuscript. Changes to the original manuscript are with “track changes”. We believe our manuscript is stronger as a result of the modifications. An itemized point-by-point response to the Reviewers’ comments is presented below. 

REVIEWER'S REPORT

Reviewer 1:

  1. Thank you for the opportunity to review this manuscript. In this paper, the authors presented systematic review was to analyse the outcome measures of functional tests and patient-reported outcomes used within an aquatic environment to identify those with a greater degree of applicability in this environment.

The manuscript is interesting and I believe that this manuscript presents novelty for publication.

In my opinion, this is a typical review using the appropriate methods for review.

  • Authors’s Answer: Thank you very much for the time and effort spent reviewing this document. We have made an itemized list responding to each of the comments made by the reviewers. We consider that thanks to the modifications made, the document has improved, making it more understandable.

  1. if I can suggest changes, I suggest that Table 2. provide reference numbers in accordance with the references, it will help you find the right article.
  • Authors’s Answer: Thank you very much for your suggestion. We have added reference numbers within table 2.

  1. If possible, it would be good to add numerical references to the articles in which the test was used in Table 3.
  • Authors’s Answer: Thank you very much for your suggestion. We have added reference numbers within table 3.

  1. In the discussion, I also suggest using numerical references in accordance with the reference list, this will allow the reader to find the right article faster.
  • Authors’s Answer: Thank you very much for your suggestion. We have added reference numbers within the discussion section.

Reviewer 2 Report

The systematic review carried out to describe the psychometric characteristics of different measuring instruments in an aquatic environment.
The introduction is adequate, the methodology follows the quality standards of a systematic review, the results are well presented, and the discussion is in line with the results. The paper will be improved if the authors add a section of the studio's limitations and recommendations more directed (in general) in the use of these measurements by clinicians.

Author Response

ITEMIZED LIST OF THE REVIEWERS COMMENTS

Manuscript ID: ijerph-853695

Assistant Editor
Mrs. Adelina Platon

We would like to thank the Reviewers for their thoughtful and constructive comments. We have considered all suggestions and have incorporated them into the revised manuscript. Changes to the original manuscript are with “track changes”. We believe our manuscript is stronger as a result of the modifications. An itemized point-by-point response to the Reviewers’ comments is presented below. 

REVIEWER'S REPORT

Reviewer 2:

  1. The systematic review carried out to describe the psychometric characteristics of different measuring instruments in an aquatic environment.

The introduction is adequate, the methodology follows the quality standards of a systematic review, the results are well presented, and the discussion is in line with the results. The paper will be improved if the authors add a section of the studio's limitations and recommendations more directed (in general) in the use of these measurements by clinicians.

  • Authors’s Answer: The authors wanted to thank the time and effort devoted to reviewing this document. We have introduced a paragraph at the end of the discussion section where limitations presented by the study have been identified. We agree with the reviewer that this paragraph is necessary in order to provide a more complete perspective when interpreting the results presented in this systematic review.

Reviewer 3 Report

A detailed review of the literature identifying tools for subjective and functional assessment of patients within an aquatic environment. This review will contribute to clinical practice immensely. Well done!  

Author Response

ITEMIZED LIST OF THE REVIEWERS COMMENTS

Manuscript ID: ijerph-853695

Assistant Editor
Mrs. Adelina Platon

We would like to thank the Reviewers for their thoughtful and constructive comments. We have considered all suggestions and have incorporated them into the revised manuscript. Changes to the original manuscript are with “track changes”. We believe our manuscript is stronger as a result of the modifications. An itemized point-by-point response to the Reviewers’ comments is presented below. 

REVIEWER'S REPORT

Reviewer 3:

  1. A detailed review of the literature identifying tools for subjective and functional assessment of patients within an aquatic environment. This review will contribute to clinical practice immensely. Well done! .
  • Authors’s Answer: The authors would like to thank the time and effort invested to correct this systematic review. Thank you very much for your comment.

Reviewer 4 Report

Good morning

It is a good article but it lacks certain details especially regarding the format to make it easier to read:

  • Do not use acronyms in the summary
  • The tables of results should be in supplementary material for easier reading of the article.
  • The tables must be redone in a homologous format: put the full text of each acronym, then the acronym Check that all the acronyms are present and in case of not having information always use the hyphen (-). Sometimes the same format is not used and not all rows are marked.
  • Put the title of the flow diagram in figure footer, check the arrows well and indicate the cause of the losses of the studies
  • In the discussion, mark the different sections in bold or italics.
  • Line 79, put Scielo in the database, there is a mistake.
  • No language bias is indicated.
  • Indicate why the cut-off score of 15 is used on the Stard Scale.
  • Mark in the text that alpha-Crombach is to be used for internal consistency.

Regards

Author Response

ITEMIZED LIST OF THE REVIEWERS COMMENTS

Manuscript ID: ijerph-853695

Assistant Editor
Mrs. Adelina Platon

We would like to thank the Reviewers for their thoughtful and constructive comments. We have considered all suggestions and have incorporated them into the revised manuscript. Changes to the original manuscript are with “track changes”. We believe our manuscript is stronger as a result of the modifications. An itemized point-by-point response to the Reviewers’ comments is presented below. 

REVIEWER'S REPORT

Reviewer 4:

  1. Good morning. It is a good article but it lacks certain details especially regarding the format to make it easier to read:
  • Authors’s Answer: Thank you very much for your comments, as well as the effort and time spent reviewing this systematic review. We have responded to all your comments as well as made the appropriate modifications. Thank you very much.

  1. Do not use acronyms in the summary
  • Authors’s Answer: Thank you very much for your suggestion. We have deleted the acronyms within the abstract.

  1. The tables of results should be in supplementary material for easier reading of the article.
  • Authors’s Answer: Thank you very much for your comment and suggestion. We have moved all tables as supplementary material at the end of the text.

  1. The tables must be redone in a homologous format: put the full text of each acronym, then the acronym Check that all the acronyms are present and in case of not having information always use the hyphen (-). Sometimes the same format is not used and not all rows are marked.
  • Authors’s Answer: thank you very much for your suggestion. We have checked all tables and we have modified them according with the reviewer’s suggestion.

  1. Put the title of the flow diagram in figure footer, check the arrows well and indicate the cause of the losses of the studies
  • Authors’s Answer: Thank you very much for your comment and suggestion. We have checked both figures, controling the lenght of the arrows, conection between them and the boxes and we have indicated the cause of the losses of the studies. .

  1. In the discussion, mark the different sections in bold or italics.
  • Authors’s Answer: Thank you very much for your comment. We have marked the different sections within the discussion in italics. .

  1. Line 79, put Scielo in the database, there is a mistake.
  • Authors’s Answer: Thank you very much for your correction. We have modified it.

  1. No language bias is indicated.
  • Authors’s Answer: Thank you very much for your comment. We have indicated within the limitation section a potential language bias.

  1. Mark in the text that alpha-Crombach is to be used for internal consistency.
  • Authors’s Answer: Thank you very much for your suggestion. We have marked within the text that alpha-Crombach was used for internal consistency.